# RegionSpot: Unleashing the power of Frozen Foundation Models for Open-World Region Understating

## Abstract

Understanding the semantics of individual regions or patches within unconstrained images, such as in open-world object detection, represents a critical yet challenging task in computer vision. Building on the success of powerful image-level vision-language (ViL) foundation models like CLIP, recent efforts have sought to harness their capabilities by either training a contrastive model from scratch with an extensive collection of region-label pairs or aligning the outputs of a detection model with image-level representations of region proposals. Despite notable progress, these approaches are plagued by computationally intensive training requirements, susceptibility to data noise, and deficiency in contextual information. To address these limitations, we explore the synergistic potential of off-the-shelf foundation models, leveraging their respective strengths in localization and semantics. We introduce a novel, generic, and efficient region recognition architecture, named `RegionSpot`, designed to integrate position-aware localization knowledge from a localization foundation model (e.g., SAM) with semantic information extracted from a ViL model (e.g., CLIP). To fully exploit pretrained knowledge while minimizing training overhead, we keep both foundation models frozen, focusing optimization efforts solely on a lightweight attention-based knowledge integration module. Through extensive experiments in the context of open-world object recognition, our `RegionSpot` demonstrates significant performance improvements over prior alternatives, while also providing substantial computational savings. For instance, training our model with 3 million data in a single day using 8 V100 GPUs. Our model outperforms GLIP-L by 2.9% in mean average precision (mAP), with an even larger margin by 13.1% for more challenging and rare categories.

## 1 Introduction

Remarkable progress has been achieved in the realm of purpose-generic image-level Vision-Language (ViL) representation learning, as exemplified by foundation models like CLIP (Radford et al., 2021) and ALIGN (Jia et al., 2021). These advancements have led to significant performance improvements across a diverse spectrum of vision and multi-modal downstream tasks (Gu et al., 2021; Zhou et al., 2022). The efficacy of these approaches can be largely attributed to their utilization of extensive datasets, typically encompassing millions, if not billions, of training samples replete with rich information. In the pursuit of a more nuanced approach to visual analysis, researchers have also ventured into the realm of universal region-level (e.g., objects) comprehension. This is evident in recent research endeavors (Gu et al., 2021; Zang et al., 2022; Ma et al., 2022; Du et al., 2022; Zhong et al., 2022; Kuo et al., 2022; Lin et al., 2022; Ma et al., 2023). A common approach to this involves learning the semantics of image regions by applying an image-level pretrained model (e.g., CLIP) to cropped regions, followed by representational distillation using the output of a detection model (Gu et al., 2021; Zang et al., 2022), as depicted in Figure 1(a). However, utilizing individual cropped regions in this design leads to the loss of crucial contextual information, which can hinder recognition performance. Kuo et al. (2022) introduced an open-world detector with a fixed ViL model, bypassing knowledge distillation. However, the use of ROIAlign (He et al., 2017) for region feature extraction poses limitations. Furthermore, directly applying an image-level

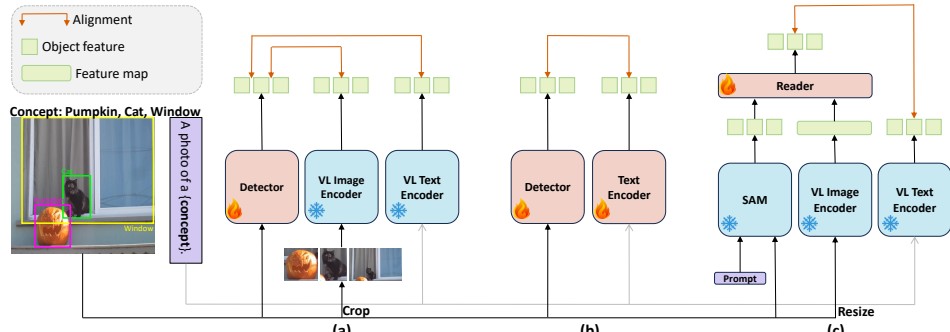

Figure 1: **Illustration of typical region-level visual understanding architecture**. (a) Learning the region recognition model by distilling image-level ViL representations from cropped regions and incorporating them into a detection model (*e.g.*, (Gu et al., 2021)). (b) Fully fine-tuning both vision and text models with a substantial dataset of region-label pairs. (c) Our proposed approach integrates pretrained (frozen) localization and ViL models, emphasizing the learning of their representational correlation.

model to isolated local regions is less effective, as the model was pretrained on entire images encompassing both object regions and surrounding context.

An alternative, albeit brute-force, approach revolves around constructing region-level representations from scratch, harnessing an extensive dataset that pairs regions with labels (Li et al., 2022; Yao et al., 2022; Zhang et al., 2022; Yao et al., 2023) (Figure 1(b)). Nevertheless, this approach grapples with challenges such as the proliferation of noisy pseudo-labels and significant training costs.

Furthermore, significant advancements have materialized in the realm of class-agnostic visual localization techniques, as illustrated by the notable work of SAM (Kirillov et al., 2023). This approach is characterized by its distinctive feature—an integration of position-aware localization knowledge, which we consider a valuable complement to the inherent capabilities of ViL models. Expanding upon this conceptual framework, our research introduces an innovative architectural paradigm at the region level, herein referred to as **RegionSpot**. This framework seamlessly incorporates large pretrained ViL and localization models within an efficient training regimen, obviating the necessity for an extensive repository of region-label pairings. Our methodology centers on the acquisition of the correlation between localization data extracted from 'local' regions by the localization model and the semantic representations encompassing the entirety of the image, derived from the ViL model. This strategic approach permits us to circumvent the conventional fine-tuning of both pre-trained models—wherein they remain 'frozen' during training—thereby safeguarding the integrity of their rich knowledge and ensuring its maximal utilization, all while mitigating the potential for performance degradation. To enact this cross-model correlation, we employ the cross-attention mechanism (Vaswani et al., 2017). In this configuration, the localization feature assumes the role of the 'query', whereas the ViL feature assumes dual roles as both the 'key' and 'value'. This implementation effectively facilitates the fusion of semantic and localization information in a manner that is amenable to learning and yields substantive efficacy.

Our **contributions** are as follows: (1) We introduce the concept of integrating off-the-shelf foundation models to tackle region-level visual understanding. (2) To achieve this objective, we introduce a novel architectural paradigm called *RegionSpot*, which does not necessitate training from scratch. This approach excels in both optimization efficiency and data utilization. By circumventing the fine-tuning of both localization and Vision-Language (ViL) components, our architecture retains its openness and adaptability, welcoming the seamless integration of advancements in both domains. Extensive experimentation in the context of open-world object understanding confirms the superior performance of our method, even with a substantially smaller number of learnable parameters. Remarkably, *RegionSpot* surpasses the state-of-the-art GLIP-L by 2.9% in mAP, with an even more substantial advantage of 13.1% observed for the more intricate rare categories.

## 2   RELATED WORK

**Zero-shot in image recognition**   Zero-shot image recognition is the task of recognizing categories that have not been seen during training. In Farhadi et al. (2009) and Jayaraman & Grauman (2014), the authors utilized visual attributes to facilitate knowledge transfer to unfamiliar categories. Re-

searchers have also investigated the utilization of class hierarchies, similarities, and object parts to enhance knowledge transfer, as demonstrated in the works of Rohrbach et al. (2011); Akata et al. (2016); Zhao et al. (2017); Xie et al. (2020).

Recent research has focused on aligning latent image-text embeddings for classifying and describing visual content. Frome et al. (2013) pioneered the establishment of a visual semantic space through deep learning. Subsequently, CLIP (Radford et al., 2021) and ALIGN (Jia et al., 2021) attained impressive results via contrastive learning with extensive collections of image-text pairs, showcasing exceptional performance across diverse benchmarks. In contrast to previous endeavors that primarily addressed image-level recognition, we focus on fine-grained recognition of visual elements at the regional level.

**Zero-shot in region understanding** In zero-shot object recognition, the aim is to enable object detectors to identify categories not encountered during training, such as (Ren et al., 2015; Sun et al., 2021; Yan et al., 2023). Researchers have explored various methods to bridge the gap between known and unknown categories using pre-trained semantic or textual features (Socher et al., 2013; Reed et al., 2016; Changpinyo et al., 2017), knowledge graphs (Salakhutdinov et al., 2011; Wang et al., 2018), and more. Inspired by the zero-shot capabilities of Vision-and-Language (ViL) like CLIP (Radford et al., 2021), several approaches have sought to integrate pretrained Vision-and-Language (ViL) models. For example, Zang et al. (2022); Gu et al. (2021); Du et al. (2022) proposed a method to distill learned image embeddings from CLIP for target detection by focusing on cropped proposal regions. Another approach, RegionCLIP (Zhong et al., 2022) employs a multistage training strategy. It starts by generating pseudo-labels from captioning data and then proceeds with region-word contrastive pretraining before transferring the knowledge to the detection task. Li et al. (2022) took a novel approach by formulating object detection as a grounding problem and incorporating additional grounding data to enhance semantic alignment at both phrase and region levels. Their results demonstrated improved performance, even on fully-supervised detection benchmarks. Yao et al. (2022) leveraged large-scale image captioning datasets and expanded their knowledge database using generated pseudo-labels, bolstering their detection capabilities. The use of generated pseudo-labels effectively extended the detectors' generalization ability.

However, these methods face computational challenges and are susceptible to training data inconsistencies and image-level distractions. Differing from these studies, we explore the synergistic benefits of foundation models SAM (Kirillov et al., 2023) and CLIP (Radford et al., 2021). Leveraging their strengths in localization and semantics, we propose an innovative region recognition framework.

## 3 METHOD

Our objective is to employ a pretrained ViL model and a localization model, trained on extensive data, to obtain region-level representations. These representations facilitate robust object conceptualization, especially for open-world region recognition. Figure 2(a) illustrates our approach, named `RegionSpot`. In the following sections, we will begin with a brief introduction to the foundational models in Section 3.1, followed by a comprehensive explanation of our approach to learning region-word alignment across two pretrained models in Section 3.2.

### 3.1 PRELIMINARIES: DIFFERENT FOUNDATION MODELS

**Vision-language foundation models** use contrastive learning to map visual and textual data into a shared embedding space through a contrastive loss. This technique, exemplified by CLIP with 400 million text-image pairs (Radford et al., 2021), and ALIGN with 1.8 billion pairs (Jia et al., 2021), aims to minimize the distances between images and their textual descriptions while maximizing distances between unrelated pairs.

**Localization foundation models** Significant strides have been made in the realm of localization foundation models. A prominent example is the pioneering SAM model (Kirillov et al., 2023), which has been trained on the extensive SA-1B dataset, boasting more than 1 billion automatically generated masks—an unprecedented scale, surpassing existing segmentation datasets by a factor of 400. This dataset also comprises 11 million images.

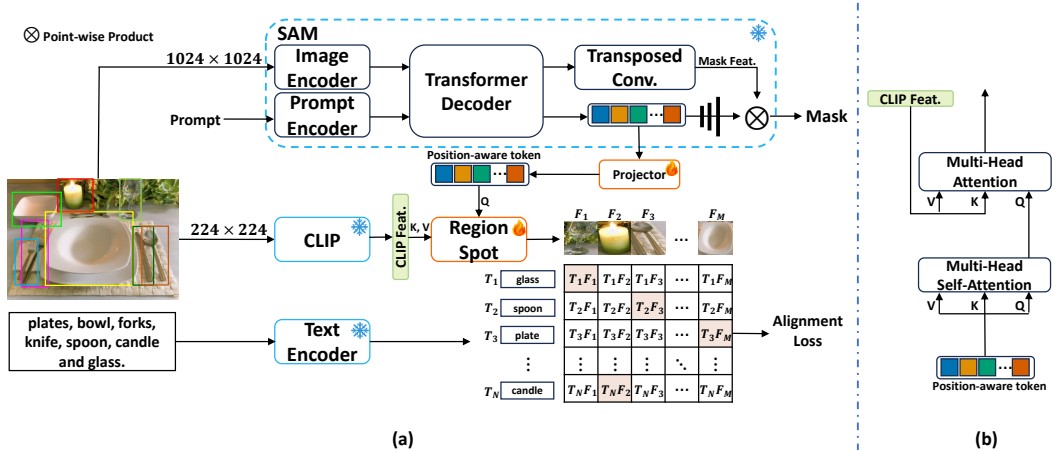

Figure 2: Overview of our proposed `RegionSpot`. (a) We integrate position-aware tokens from a localization model, such as SAM, with image-level feature maps extracted from a ViL model like CLIP. This integration yields region-level semantic tokens, which are then subjected to region text alignment. (b) Our cross-modal feature interaction design based on the attention mechanism.

SAM comprises three core modules: (a) Image encoder: Utilizing a ViT-based backbone, this module extracts image features, yielding image embeddings. (b) Prompt encoder: It encodes positional information from input points, boxes, or masks to facilitate the mask decoder. (c) Mask decoder: This transformer-based decoder leverages both the extracted image embeddings and prompt tokens to make final mask predictions. One of SAM's remarkable features is its robust zero-shot generalization to novel data, obviating the need for domain-specific fine-tuning. Thanks to extensive training on a vast repository of prompt-text pairs, SAM demonstrates exceptional proficiency in object localization.

## 3.2 REGION TEXT ALIGNMENTS WITH FROZEN FOUNDATION MODELS

In this section, we describe how we extract position-aware tokens from the localization foundation model and generate image-level semantic features using the ViL foundation model. We achieve inter-model association through a cross-attention mechanism that facilitates region text alignment.

**Region-level position-aware tokens** In our approach, we utilize manually-annotated object bounding boxes, denoted as $R = \{r_i\}, i = 1, .., N$, as regions of interest in the images. For each of these regions, represented as $R$, we extract position-aware tokens using the SAM model, denoted as $P = \{p_i\}, i = 1, .., N$.

As depicted in Figure 2, SAM employs a mask decoder to generate a mask based on a provided prompt. This process utilizes a transformer decoder, similar to the architecture in DETR (Carion et al., 2020), to generate an object token. This object token plays a crucial role in predicting the prompt mask, subsequently predicting dynamic MLP weights and performing a point-wise product with the mask features. We refer to this resulting token as "position-aware" because it encodes essential information about the object, including details about its texture and position. Following this, a projector is applied to map the output dimension of the position-aware token to the image-level feature space as discussed below.

**Image-level semantic feature maps** A single image can encompass multiple objects across numerous categories, capturing integrated context. We can conceptually view an image's feature map as a composition of region embeddings with varying structures. To fully capitalize the ViL model, we resize the input image to the required dimensions without cropping. Subsequently, we input this resized image into the ViL model, yielding the image-level semantic feature map denoted as $V$.

**Relating position-aware tokens and semantic feature maps** Our model, referred to as `RegionSpot`, efficiently establishes connections between region-level position-aware tokens and

image-level semantic feature maps using the cross-attention mechanism (Vaswani et al., 2017). In this mechanism, position-aware tokens $P$ serve as queries, while semantic feature maps $V$ take on the roles of both keys and values. This relationship is formulated as follows:

$$S = \text{Softmax}\left(\frac{F_p K_v^T}{\sqrt{C}}\right) V_v,  \tag{1}$$

where $F_p$ represents a transformation of $P$, $K_v$ and $V_v$ are derived from separate linear projections of $V$, and $C$ is the projected feature dimension. This approach, well-established in the literature, has consistently demonstrated its effectiveness in information fusion. In our work, we extend its application to enhance region-level understanding in open-world scenarios. Specifically, we leverage this mechanism to integrate positional information with semantic content extracted from two distinct models at the regional level, while also considering the broader context from the entire image, as depicted in Figure 2(b).

**Loss function**    In line with prior research, we generate text embeddings by processing category-specific texts along with prompt templates, like *a photo of category in the scene*, using the text encoder. Subsequently, we perform a dot product operation between each semantic token and its corresponding text features to calculate matching scores. These scores can be supervised using the focal loss (Lin et al., 2017).

## 4    EXPERIMENTS

**Training data**    In pursuit of a robust training environment, we combined diverse datasets with varying label spaces. Our model's flexible architecture allowed us to seamlessly replace one-hot labels with class name strings. For training, we utilized publicly available detection datasets, comprising a total of approximately 3 million images. These datasets include Objects 365 (O365) (Shao et al., 2019), OpenImages (OI) (Krasin et al., 2017), and V3Det (V3D) (Wang et al., 2023), each contributing uniquely to the diverse repository.

- Objects 365 (O365) is a large-scale object detection dataset featuring 365 distinct object categories across 0.66 million images. Our research employs an enriched version with over 10 million bounding boxes, averaging approximately 15.8 object annotations per image.

- OpenImages (OI) currently stands as the largest public object detection dataset, encompassing about 14.6 million bounding box annotations, equivalent to around 8 annotations per image.

- V3Det (V3D) distinguishes itself through a hierarchical organization, meticulously structuring up to 13,029 categories within a category tree.

**Benchmark settings**    In our rigorous evaluation process, we utilized the extensive LVIS detection dataset (Gupta et al., 2019), which encompasses 1203 categories and 19809 images reserved for validation. We do not prioritize the performance on COCO (Lin et al., 2014) which includes only 80 common categories covered by the Objects365 training dataset (Shao et al., 2019). This limitation may not adequately assess a model's generalization in an open-world setting.

Since our current emphasis is not on object localization, we utilized ground-truth and class-agnostic bounding boxes from an existing detector to predict categories based on corresponding text descriptions, following the RegionCLIP approach (Zhong et al., 2022). Mean Average Precision (mAP) served as our evaluation metric.

**Implementation details**    We train `RegionSpot` using AdamW (Kingma & Ba, 2014) optimizer with the initial learning rate as $2.5 \times 10^{-5}$. All models are trained with a mini-batch size 16 on 8 GPUs. The default training schedule is 450K iterations, with the learning rate divided by 10 at 350K and 420K iterations. The training process unfolds in two sequential stages: (1) a warm-up phase leveraging the Objects365 to initiate the learning of region-word alignments, and (2) a phase of advanced learning for region-word alignments, utilizing a rich compilation from three diverse object detection datasets. The model is trained for 450K iterations at each stage.

We consider a couple of model variants: (1) `RegionSpot-BB`: This configuration integrates the base versions of both SAM and CLIP. (2) `RegionSpot-BL`: In this variant, we combine the SAM base with the more extensive CLIP large architecture.

| Method | Training data | Proposals | $AP_r$ | $AP_c$ | $AP_f$ | $AP_{all}$ |
|---|---|---|---|---|---|---|
| CLIP w/ box | - | GT | 40.6 | 53.1 | 59.2 | 48.7 |
| CLIP w/ mask | - | GT | 40.8 | 53.5 | 59.6 | 49.2 |
| RegionCLIP | CC3M | GT | 50.1 | 50.1 | 51.7 | 50.7 |
| RegionSpot-BB | O365, OI, V3D | GT | 42.0 | 45.9 | 65.6 | 53.0 |
| RegionSpot-BL | O365, OI, V3D | GT | **50.6** | **50.2** | **68.8** | **56.6** |
| RegionCLIP | CC3M | RPN | 13.8 | 12.1 | 9.4 | 11.3 |
| RegionSpot-BB | O365, OI, V3D | RPN | 10.9 | 10.2 | 13.6 | 11.8 |
| RegionSpot-BL | O365, OI, V3D | RPN | 12.7 | **13.1** | **15.7** | **14.1** |
| GLIP-T (A) | O365 | GLIP-T(B) | 6.0 | 8.0 | 19.4 | 12.3 |
| GLIP-T (B) | O365 | GLIP-T(B) | 4.2 | 7.6 | 18.6 | 11.3 |
| GLIP-T (C) | O365, GoldG | GLIP-T(B) | 7.5 | 11.6 | 26.1 | 16.5 |
| GLIP-T | O365,GoldG,Cap4M | GLIP-T | 10.1 | 12.5 | 25.5 | 17.2 |
| GLIP-L | FourODs,GoldG,Cap24M | GLIP-L | 17.1 | 23.3 | 35.4 | 26.9 |
| RegionSpot-BL | O365 | GLIP-T(B) | 12.7 | 13.1 | 15.7 | 14.1 |
| RegionSpot-BB | O365, OI, V3D | GLIP-T | 20.0 | 18.7 | 24.2 | 21.1 |
| RegionSpot-BL | O365, OI, V3D | GLIP-T | 24.9 | 21.6 | 25.5 | 23.7 |
| RegionSpot-BL | O365, OI, V3D | GLIP-L | **30.2** | **28.3** | **30.0** | **29.8** |

Table 1: Open-world zero-shot object recognition under the settings of using ground-truth (GT) boxes, RPN boxes and GLIP boxes on LVIS dataset.

## 4.1 Zero-shot Inference for Region Recognition

**Setup**  In our evaluation process, we utilize zero-shot region recognition to probe the capability of our model in aligning regions with textual descriptions. Recognizing that our primary focus isn't on object localization, we rely on a pre-existing proposal generator for best result. The model's responsibility is to determine categories based on the accompanying texts. Our exploration spans three specific types of region proposals: (1) Manually-annotated Object bounding boxes: Utilizing manually-annotated objects bounding boxes as region proposals, this methodology is designed to measure the aptitudes of model in object recognition. By doing so, we effectively sidestep potential errors arising from localization. (2) Region Proposals from RPN: In this scenario, region proposals are sourced from a Region Proposal Network (Girshick, 2015) (RPN), following the approach delineated in RegionCLIP (Zhong et al., 2022). The perfomance in this context is contingent upon both the precision localization of RPN and the inherent of RegionSpot ability to recognize objects. (3) Region Proposals from a ViL Detector: To maintain a level playing field with the Vision-and-Language (ViL) detector, we employ their predicted boxes. Subsequently, we predict its class to thoroughly evaluate our model in this context. Collectively, these distinct methodologies offer a multifaceted view into the model's proficiency, ensuring a comprehensive assessment of its performance under varied conditions.

**Baselines**  We consider two baselines: (1) Direct Region Cropping: For this method, we simply extract a region from the image using the specified bounding box. The cropped region is then fed into the model for region-level visual recognition. (2) Box-Prompted SAM Masking: In this approach, we utilize the bounding box as a prompt for the SAM. This produces a mask that not include noisy background, which is then used to crop a specific region from the image. Subsequently, this cropped region is input into the model for region-level visual recognition.

**Results**  Results on the LVIS benchmark are presented in Table 1. With ground-truth bounding boxes as region proposals, our model substantially surpasses the CLIP baselines (which applies CLIP on image crops) by a large margin (*e.g.*, **48.7, 49.2** *vs.***56.6**). Our method also shows a pronounced advantage over RegionCLIP (*e.g.*, **50.7** *vs.***56.6**), despite RegionCLIP is fully fine-tuned on CC3M. These promising results suggest that our method effectively improves the visual recognition ability of CLIP on image regions. Moreover, in simulation of real-world cases, we move forward to test our method with noisy region proposals generated from off-the-shelf RPN. It can be seen that RegionSpot still consistently outperforms RegionGLIP (*e.g.*, **11.3** *vs.***14.1** on $AP_{all}$) in this case, demonstrating the robustness of our method.

From the above experiments, we notice a significant performance drop when switching from ground-truth to noisy region proposals. This is mainly due to the low recall of off-the-shelf RPN on LVIS objects. To fully exploit the potential of our method and synergy with the advancements of open world object detection (OWD), we further utilize region proposals from state-of-the-art OWD models, *i.e.*, GLIP, as our region prompts. Comparing with GLIP-T trained solely on the objects365 dataset, we can observe a considerable performance gain achieved by RegionSpot (*e.g.*, **4.2** *vs.***12.7** on $AP_r$ and **11.3** AP *vs.***14.1** on $AP_{all}$). After scaling up the training data, our models maintains superior performances over their GLIP counterparts. For instance, RegionSpot surpasses GLIP-T by **14.8** $AP_r$ with less training data, showing compelling scaling behavior with data scale. For more extensive evaluation, we also utilize bounding boxes generated by GLIP-L as prompts. It is noteworthy that RegionSpot achieves an impressive **13.1** increase in $AP_r$ compared to GLIP-L, even when trained on less data at higher efficiency.

| Method | Backbone | Training data | Training time (GPU hours) | Learnable Parameters (M) |
|---|---|---|---|---|
| RegionCLIP | ResNet50×4 | CC3M | 4.6K | - |
| GLIP-T | Swin-T | O365, GoldG, Cap4M | 92.1K | - |
| GLIP-L | Swin-L | FourODs,GoldG,Cap24M | 120K | 289 |
| RegionSpot | ViT-B, ViT-L | O365, OI, V3D | **0.2K** | 35 |

Table 2: Comparisons with the training efficiency.

**Training efficiency.** To illustrate the training efficiency of `RegionSpot`, we benchmark its GPU training hours against RegionCLIP and GLIP, as showcased in Table 2. Even though we utilize the ViT-Large as our backbone, our model achieves faster training. This efficiency can be attributed to our frozen approach and processing of images at a reduced resolution of 224x224 for the CLIP Large. All benchmarks were executed in a consistent hardware environment, leveraging eight NVIDIA V100 GPUs. In stark contrast, GLIP necessitates an extensive 92K GPU hours, a whopping 436 times more than our approach, mainly due to its exhaustive fine-tuning of both Vision and Text models. Interestingly, even when RegionCLIP adopts a smaller backbone akin to ours, it still requires 4.6K GPU hours.

## 4.2 ABLATION STUDY

We conducted an ablation study for `RegionSpot-BL` using the boxes generated by GLIP. Unless otherwise mentioned, training was performed on three different detection datasets.

| Data | $AP_r$ | $AP_c$ | $AP_f$ | AP |
|---|---|---|---|---|
| Objects365 | 11.9 | 13.6 | 20.2 | 15.9 |
| Objects365 + OpenImages | 16.4 | 18.2 | 23.7 | 20.1 |
| Objects365 + OpenImages + V3DET | **24.9** | **21.6** | **25.5** | **23.7** |

Table 3: Effect of increasing the detection training data.

**Benefit of increasing the detection training data** Table 3 showcases the performance improvements observed when augmenting the training data size. Through our proposed framework, integrating additional detection data from diverse sources consistently enhances the capabilities rooted in pretrained knowledge. Compared to training with only Objects365, including OpenImages effectively improves the overall AP from 15.9 to 20.1. The inclusion of V3Det further propels the performance, achieving an impressive overall AP of 23.7. This improvement is particularly significant for rare categories, with an increase from 16.4 to 24.9 (a gain of +8.5 AP), attributable to its extensive vocabulary.

**Enhancement with CLIP vision embedding** We conjugate that a key ingredient with `RegionSpot` is the use of semantic information from CLIP vision encoder. To validate this assertion, we began our evaluation without the CLIP feature and subsequently integrated the class token output from the CLIP vision encoder. Results in Table 4a demonstrate that: (1) The CLIP feature offers a significant boost, pushing the baseline without CLIP vision encoder to **22.1**, which

| CLIP | AP | Position-aware tokens | AP | Depth | AP |
|---|---|---|---|---|---|
| w/o CLIP vision. | 8.0 | Prompt encoder | 18.6 | 1 | 23.2 |
| + CLIP feat. map | 22.1 | Transformer | **23.7** | 3 | **23.7** |
| + Class token | **23.7** | MLP | 20.4 | 6 | 22.8 |
| (a) | | (b) | | (c) | |

Table 4: **Ablation experiments on LVIS. (a)** The effective of CLIP vision encoder; **(b)** Position-aware tokens selection ; **(c)** Depth of `RegionSpot`.

suggests inherent semantic limitations in SAM. (2) More notably, the inclusion of CLIP enhances overall performance to **23.7**. This underscores the potential of the class token to encapsulate global information from the entire image.

**Position-aware tokens selection in SAM**    The position-aware tokens are generated by intermediary module in the SAM. We examined various locations for this generation, specifically after the Prompt encoder, the Transformer decoder, and the MLP within the SAM. Results presented in Table 9b indicate that generating output tokens after the Transformer decoder yields the best performance. This observation is expected since tokens derived from the Prompt encoder are relatively undeveloped. Surprisingly, it can outperform GLIP (*i.e.*, **18.6** vs. **17.2**). Moreover, there is a performance decline after the MLP, which can be attributed to dimensional reduction.

**Module architecture**    Another pivotal aspect of `RegionSpot` is the depth of model. To assess its impact, we experimented by varying the depth of our model. As indicated in Table 4c, it is imperative for the model to have a sufficiently large depth, such as 3 blocks, without being excessive.

| Prompt | $AP_r$ | $AP_c$ | $AP_f$ | AP |
|---|---|---|---|---|
| baseline | 19.6 | 15.1 | 21.2 | 18.5 |
| w/ mutiple boxes prompt | 23.2 | 19.2 | 25.0 | 22.1 |
| w/ text prompt | **24.9** | **21.6** | **25.5** | **23.7** |

Table 5: Ablation study on prompt engineering.

**Prompt engineering**    Finally, we carried out an ablation study focusing on prompt engineering, incorporating both box prompts in SAM and text prompts in the text encoder. As evidenced by the results in Table 5: (1) Leveraging multiple boxes as prompts in SAM boosts performance, achieving an AP of 22.1. This enhancement is credited to the self-attention mechanism of `RegionSpot`, which adeptly fuses information from varied regions. (2) Further utilizing text prompts results in a modest performance boost, specifically an increase of 1.6 AP.

## 4.3    VISUALIZATION

**Result visualization**    In Figure 3, we present the results of bounding region recognition on the LVIS  (Gupta et al., 2019) dataset, comparing between GLIP and `RegionSpot`. To assess the zero-shot recognition capability, we employ the same bounding boxes for both models. As observed, `RegionSpot` can distinguish even subtle differences, recognizing smaller objects like "lemon" and "tennis ball" and similar objects like "lantern" and "fireplug". Notably, `RegionSpot` stands out in terms of the accuracy of its label predictions, especially within the category of rare classes.

**Model behavior visualization**    To gain more intuitive understanding on the effect brought by our `RegionSpot`, we examine the cross attention map on LVIS (Gupta et al., 2019). We take the output tokens as 'query' and CLIP feature map as 'key' and 'value'. For clearer visualization, we omit the class token from the CLIP semantic feature. The resulting attention map clearly depicts the correspondence between the position-aware tokens generated by SAM and the feature map produced by CLIP. Using this arrangement, we gain a visual insight into how `RegionSpot` establishes connections between distinct features. As depicted in Figure 4, the attention maps vividly showcase `RegionSpot` capability to seamlessly incorporate both SAM and CLIP. Such visualizations serve a dual purpose: they highlight the efficacy of our method, and simultaneously, shed light on the intricate mechanisms underpinning the `RegionSpot`.

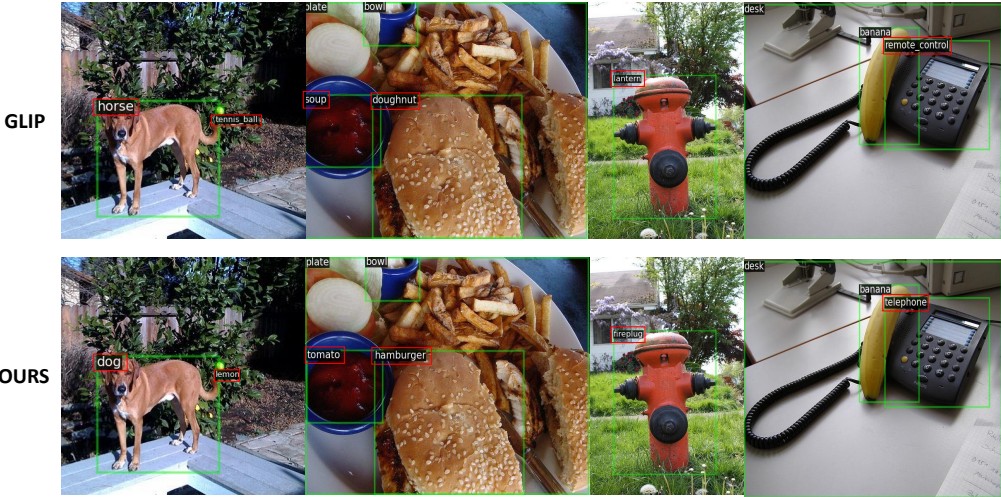

Figure 3: Qualitative prediction results for the LVIS dataset (Gupta et al., 2019) are shown for GLIP-T (Li et al., 2022) (first row) and `RegionSpot` (second row). Our model recognizes the objects more accurately. Best viewed with zooming-in.

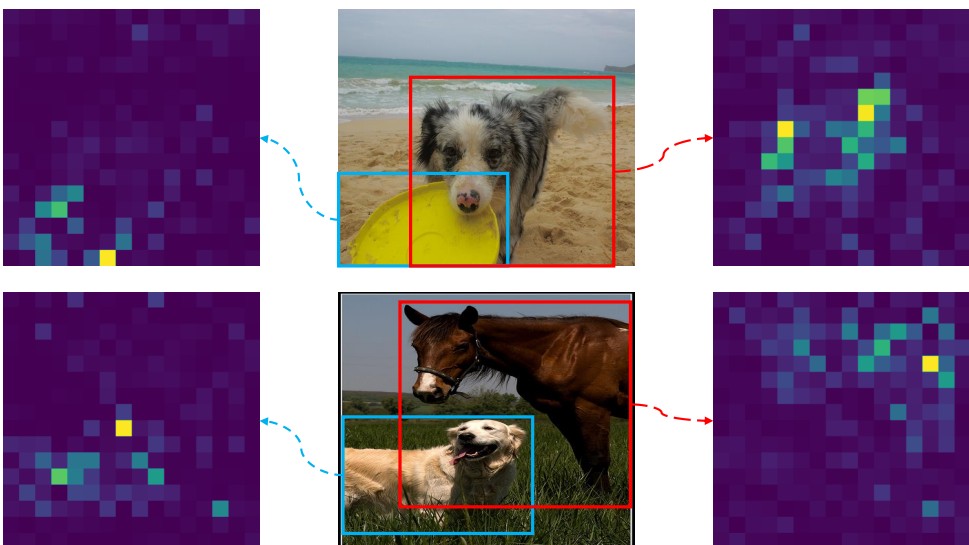

Figure 4: Cross-attention map from `RegionSpot`. These maps demonstrate that the position-aware token aligns effectively with the semantic feature map of the entire image. In each row, the blue and red boxes are corresponding to the left and right maps respectively.

## 5 CONCLUSION

In this study, we introduce `RegionSpot`, a novel and efficient framework leveraging frozen vision and vision-language foundation models for region recognition, eliminating the need for training from scratch. To fully exploit knowledge in pretrained models and minimize the training overhead, we keep both foundation models frozen and focus optimization efforts solely on a lightweight attention-based knowledge integration module. Extensive experiments in the context of open-world object understanding confirms the superior performance of our method, even with a substantially smaller number of learnable parameters, which distinguishes our method and enables efficient training. Impressively, *RegionSpot* outperforms the leading GLIP-L by 2.9% in mAP, and this lead grows to 13.1% when considering complex rare categories.

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

## A    ADDITIONAL EXPERIMENTAL RESULTS

**Zero-shot instance segmentation**    We evaluate the performance of instance segmentation in a zero-shot setting using the LVIS dataset (Gupta et al., 2019). By leveraging the output from GLIP as the prompt, we direct it to `RegionSpot`for mask and class prediction. The mask AP is evaluated using the released X-Decoder (Zou et al., 2023) and OpenSeeD (Zhang et al., 2023), both of which are trained with mask-text pairs. Impressively, as indicated in Table 6, `RegionSpot` outstrips X-Decoder and OpenSeeD by margins of 14.1% and 3.9%, respectively. These outcomes suggest that our proposed `RegionSpot` can effectively harness the foundational model's capabilities to achieve more accurate region understanding.

| Method | $AP_r$ | $AP_c$ | $AP_f$ | AP |
|---|---|---|---|---|
| X-Decoder | - | - | - | 9.4 |
| OpenSeed | - | - | - | 19.6 |
| `RegionSpot-BL` | **21.5** | **25.0** | **23.2** | **23.5** |

Table 6: Evaluation of zero-shot instance segmentation on the LVIS minival dataset.

**Zero-shot object detection on LVIS minival5k**    To fully exploit the potential of our method, we report on MiniVal containing 5,000 images introduced in MDETR (Kamath et al., 2021). We use the output proposals from GLIP as the prompt. As shown in the Table 7, although we use less 9x training data, our models maintains superior performances over their GLIP-L by 5.0 APr on the MiniVal. Further, our method also surpasses Grounding DINO-L 11.0 APr on the MiniVal, which used the advanced detector.

| Method | Training Data | $AP_r$ | AP |
|---|---|---|---|
| GLIP-L | FourODs,GoldG,Cap24M | 28.2 | **37.3** |
| GroundingDINO-L | O365,OI,GoldG,Cap4M,COCO,RefC | 22.2 | 33.9 |
| `RegionSpot-BL` | O365,OI,V3DET | **33.2** | 36.9 |

Table 7: Evaluation of zero-shot object detection on the LVIS minival dataset.

**More results under ViLD-protocal**    To thoroughly evaluate our method, we conducted experiments using the ViLD protocol (Gu et al., 2021), training on base categories and testing on novel ones with the LVIS AP metric. We adapted our training to the LVIS-base dataset. As shown in Table 8, RegionSpot demonstrates competitive performance. It outperforms the similarly frozen-backbone F-VLM by 1.1 $AP_r$ with RPN proposals. With GLIP-T proposals, RegionSpot achieves an AP of 20.4%. When we compared to RegionCLIP, which benefits from additional caption pretraining, RegionSpot significantly outperforms the pretrained version of RegionCLIP by 2.6% when utilizing RPN as the proposal generator.

| Method | Proposals | $AP_r$ |
|---|---|---|
| ViLD | RPN | 16.1 |
| RegionCLIP | RPN | 17.1 |
| Detic-ViLD | RPN | 17.8 |
| F-VLM | RPN | 18.6 |
| RegionSpot | RPN | 19.7 |
| RegionSpot | GLIP-T | **20.4** |

Table 8: Comparison under the ViLD protocol Gu et al. (2021). All methods use the ResNet50 backbone.

**Ablation study of SAM-CLIP Pipline**    We posit that RegionSpot effectively leverages the pretrained knowledge from foundational models. To evaluate its capabilities in region understanding,

we compare it with several alternatives that integrate SAM and CLIP: all baseline use SAM segment everything mode to generate the proposal. (1) Directly use the Frozen CLIP to Tag. (2) Utilizing ROI Align to obtain region features and then applying a simple learnable projector to adapt these features. We assess each model using SAM and GLIP proposal as prompt, respectively. As shown in Table 9a, RegionSpot significantly outperforms all the baseline models.

**Ablation study of SAM model**     We conjugate that a key ingredient is the position-aware information from SAM. We evaluating the impact of different SAM model sizes, such as ViT-L, is essential. We conducted experiments with varying SAM model sizes. As shown in the Table 9b, our findings are summarized as follows: (1) Impact of SAM Model Size: Our results indicate that the use of larger SAM models (e.g., SAM-L) improves mask AP due to the higher quality of mask generation. However, for box AP, the improvement is not significant. This is because the SAM mask token primarily contributes position-aware knowledge, which is already sufficiently captured by ViT-B and ViT-L. (2) Choice of SAM Model: Given our focus on region recognition, we opted for SAM-B, balancing performance and computational efficiency.

| method | Proposals | $AP_r$ |
|---|---|---|
| SAM-CLIP | SAM | 8.6 |
| SAM-CLIP w/projector | SAM | 9.6 |
| RegionSpot-BB | SAM | 10.6 |
| RegionSpot-BB | GLIP | 20.0 |

(a)

| SAM | box $AP_r$ | mask $AP_r$ |
|---|---|---|
| ViT-B | 24.9 | 22.8 |
| ViT-L | 24.7 | 23.6 |

(b)

Table 9: **Ablation experiments on LVIS. (a)** The effective of RegionSpot; **(b)** The effective of SAM

# B  MORE VISUALIZATIONS ON LVIS

Figure 5 provides more examples on LVIS (Gupta et al., 2019).

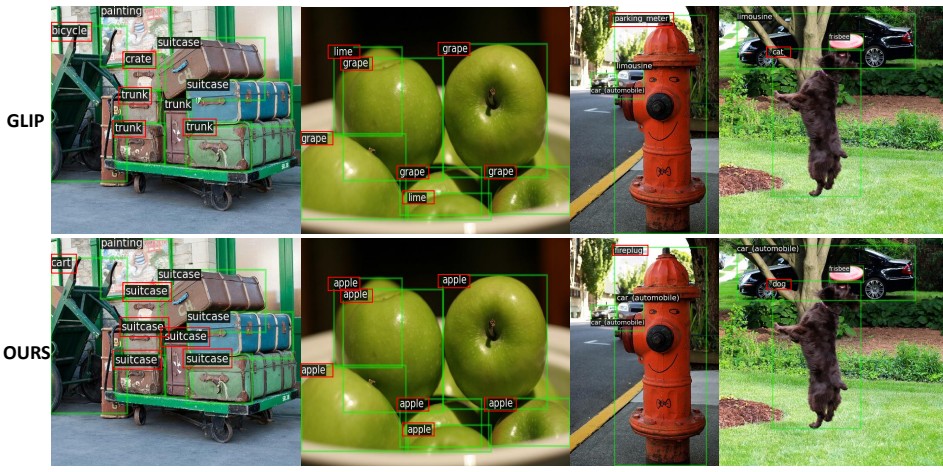

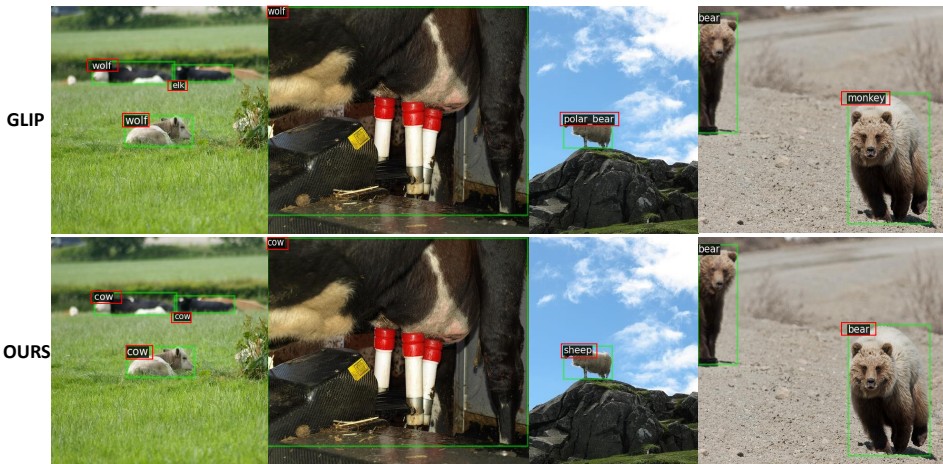

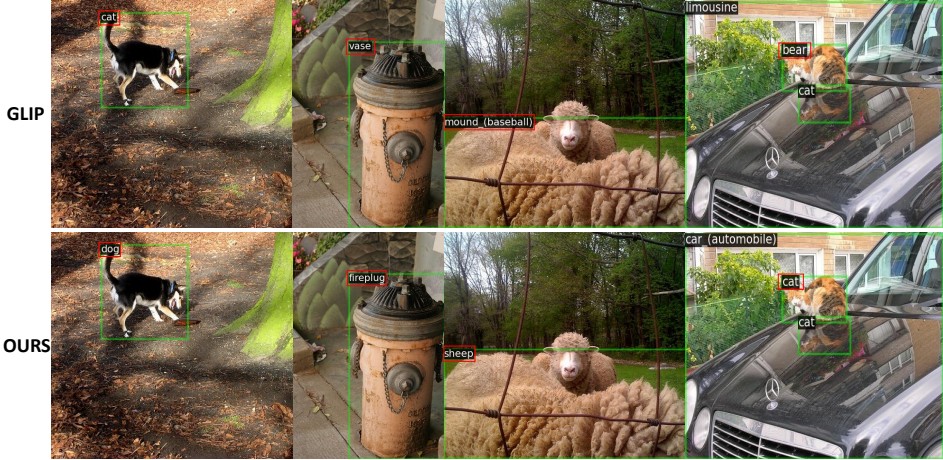

Figure 5: More visualizations in comparison with GLIP. Best viewed when zoomed-in.

