# OpenReview forum: "RegionSpot: Unleashing the Power of Frozen Foundation Models for Open-World Region Understanding"
_ICLR.cc/2024/Conference — Submitted to ICLR 2024_

### Official Review · Reviewer_3vj6 · 2023-10-30

**Soundness:** 3 good
**Presentation:** 3 good
**Contribution:** 2 fair
**Rating:** 3
**Confidence:** 4

**Summary:**

The paper aims to combine SAM and CLIP to improve region-level visual understanding. Specifically, the proposed RegionSpot method freezes the whole SAM and CLIP models, and adds new layers to them to let the position-aware tokens from SAM interact with the image-level features from CLIP, thus leading to region-level semantic tokens. After training the new layers on Object365, OpenImages, and V3D, the method shows decent performance on LVIS open-world object recognition.

**Strengths:**

- The motivation of combining the capabilities of SAM and CLIP is clear and makes sense.
- The method is simple and easy to implement, as most of the parameters are frozen and only several new layers are trained.

**Weaknesses:**

- The claim of 6.5% and 14.8% improvements over GLIP makes no sense. The comparison is not fair. The paper only compares with GLIP-Tiny, which is way smaller than the proposed RegionSpot. GLIP-Tiny uses Swin-Tiny which has 29M parameters, while RegionSpot-BB has at least 160M parameters. According to the GLIP paper, GLIP-L achieves 26.9 AP on LVIS, which is better than the best RegionSpot-BL's 23.7 AP.
- Evaluation is limited. The paper only tests on open-world LVIS recognition. It would be more convincing to do a more comprehensive evaluation.
- It is not appropriate to claim a zero-shot recognition on LVIS, as RegionSpot is trained on the combination of Object365, OpenImages, and V3D, which shares a lot of common categories, objects, and scenes with LVIS.
- Please check the reference carefully. For example, " However, the use of ROIAlign (Ren et al., 2015) for region feature extraction...". The ROIAlign is not proposed in Ren et al., 2015.

**Questions:**

- The paper only experiments with different CLIP models, i.e., CLIP-base and large. What about using larger SAM models, e.g., RegionSpot-LL?
- More details can be reported, e.g., the number of new parameters.

---

> ### Author Response · Authors · 2023-11-20
> **Official Response to Reviewer 3vj6**
>
> Thank you for the helpful review. We carefully address your questions and comments below and have updated the submission pdf accordingly.
>
> > The claim of 6.5% and 14.8% improvements over GLIP makes no sense. The comparison is not fair. The paper only compares with GLIP-Tiny, which is way smaller than the proposed RegionSpot. GLIP-Tiny uses Swin-Tiny which has 29M parameters, while RegionSpot-BB has at least 160M parameters. According to the GLIP paper, GLIP-L achieves 26.9 AP on LVIS, which is better than the best RegionSpot-BL 23.7 AP.
>
> Thanks for pointing out this. To address this question, we have now provided extra comparison under GLIP protocol. This table below shows that our method is superior over GLIP-L despite using less training data (3M vs 27M) and less learnable parameters (35M vs. 289M). Both conditions are related to the training cost directly, which means our training is significantly more efficient.
>
> Table 1: Comparison under GLIP protocol.
> | Method | Training Data | Data Size | Fully finetune | Training Time | MiniValAPr | MiniValAPall | Val APr | Val APall |
> | --- | --- | --- | --- | --- | --- | --- | --- | --- |
> | GLIP-L | FourODs,GoldG,Cap24M | 27M | Yes | 120K | 28.2 | 37.3 | 17.1 | 26.9 |
> | RegionSpot-BL | O365,OI,V3DET | 3M | No | 0.2k | 33.2 | 36.9 | 30.2 | 29.8 |
>
>
> > Evaluation is limited. The paper only tests on open-world LVIS recognition. It would be more convincing to do a more comprehensive evaluation.
>
> Thanks. For more comprehensive evaluation, we also perform experiments under the VILD protocol. The results, presented in Table 2 (Response to all reviewers), show that our method achieves a 19.7% APr, surpassing F-VLM 18.6% APr by 1.1% APr. Notably, F-VLM is also based on a Frozen Foundation model. This comparison demonstrates our model capability to effectively leverage pre-trained knowledge.
>
> > It is not appropriate to claim a zero-shot recognition on LVIS, as RegionSpot is trained on the combination of Object365, OpenImages, and V3D, which shares a lot of common categories, objects, and scenes with LVIS.
>
> Great point. It's true that recent large models, such as CLIP, are typically trained on vast datasets that encompass a wide range of concepts. Consequently, when these models are utilized for a downstream task, they often encounter concepts that were already part of their training data, though it is unclear which specific concepts. While this may not strictly adhere to the definition of zero-shot learning, it is widely acknowledged and applied in practice. Our setup aligns with GLIP in the utilization of extensive training data, a practice that is not novel but well-established.
> Additionally, we extend our evaluation using the protocol established by VILD, ensuring a robust and comprehensive assessment of our method. This further demonstrates our method can fully unleash pretrained knowledge.
>
> > Please check the reference carefully. For example, " However, the use of ROIAlign (Ren et al., 2015) for region feature extraction...". The ROIAlign is not proposed in Ren et al., 2015.
>
> Thanks, we will fix all in the revision.
>
> > The paper only experiments with different CLIP models, i.e., CLIP-based and large. What about using larger SAM models, e.g., RegionSpot-LL ?
>
> Thanks for this great suggestion. To address this, we have conducted additional experiments with varying SAM model backbones. Our findings are summarized as follows:
> 1. Role of SAM in Position-Aware Knowledge and Mask Generation: Our results indicate that the use of larger SAM models (e.g., SAM-L) improves mask AP due to the higher quality of mask generation. However, for box AP, there is even some slight drop in the improvement. This is because the SAM mask token primarily contributes position-aware knowledge, which is already sufficiently captured by either ViT-B or ViT-L.
> 2. Choice of SAM Model: Given our focus on region recognition, we opted for SAM-B, balancing performance and computational efficiency.
>
> | SAM-Model | Box AP_rare | Mask AP_rare |
> | --- | --- | --- |
> | ViT-B | 24.9 | 22.8 |
> | ViT-L | 24.7 | 23.6 |
>
> > More details can be reported, e.g., the number of new parameters.
>
> Great point! As suggested, we provide the number of learnable parameters as following:
> |  | Leanble Parameter(M) |
> | --- | --- |
> | RegionSpot-BB | 22 |
> | RegionSpot-BL | 35 |
>
> We hope the above information addresses all the concerns and provides further insight into our work.

---

> ### Author Response · Authors · 2023-11-22
> **Last discussion request before the discussion period ends**
>
> Dear Reviewer，
>
> Thank you for dedicating your time and expertise to review our paper and for participating in the rebuttal process. Your feedback has been instrumental in enhancing the quality of our work.
>
> We hope that our responses in the rebuttal have satisfactorily addressed your concerns. If there are any remaining issues or new points of discussion, we are fully prepared to engage in further dialogue. Given the constraints of the review timeline, we would greatly appreciate it if you could review our revised responses at your earliest convenience.
>
> Should you find that our revisions and clarifications have resolved the initial concerns, we would be grateful for your reconsideration of the initial rating. However, if there are still aspects that require clarification, we welcome the opportunity to discuss them in the remaining time.
>
> We sincerely appreciate your time and thoughtful consideration.

---

### Official Review · Reviewer_Tjjg · 2023-10-31

**Soundness:** 3 good
**Presentation:** 2 fair
**Contribution:** 2 fair
**Rating:** 5
**Confidence:** 4

**Summary:**

This paper introduces a open-world region recognition architecture, named RegionSpot, designed to integrate position-aware localization knowledge from SAM and CLIP.

**Strengths:**

1. The method performs open world object detection through pre-trained vision foundation model SAM and CLIP. SAM is a good foundation model with excellent performance. This work sucessfully adopts SAM into object detection (requiring object recognition). This idea is novel and good. Introducing vision foundation models into new tasks is an important thing, I think.
2. The experimental results are good.

**Weaknesses:**

Most of the problems come from the experiment part.
1. This method still requires existinig region proposal generation models, like ground-truth, RPN, GLIP, which makes this method extremely limited. From this view, this method is even not complete, since you cannot find such a region understanding task in reality. In addition, the pre-extracted regions are also meaningless, since SAM can also perform the region proposal generation task. Therefore, the main experimental setting is meaningless and unreasonable.
2. There are also many alternatives to perform the so-called region understanding task with SAM and CLIP. For example, SAM can directly extract region proposals and CLIP (or RegionCLIP) can predict the category tags of them. We can also add some projection layers in this way, freeze most parameters and finetuning a small part of parameters for efficient training. The author should perform experiments to compare with baselines like this. Otherwise, the effect of the position-aware localization cannot be seen.
3. The experiment in Table 1 is also unfair. RegionCLIP simply performs image-text pre-training, without object detection finetuning. However, RegionSpot performs something about detection training. Therefore, the author should finetuning RegionCLIP on the same detection datasets for a fair comparison.
4. The author should also compare with some more recent methods, like GLIP v2, Grounding DINO and so on.

**Questions:**

Most of the problems come from the experiment section. The author should provide more additional results to make the paper accepted,

---

> ### Author Response · Authors · 2023-11-20
> **Official Response to Reviewer Tjjg**
>
> Thank you for the helpful review. We carefully address your questions and comments below and have updated the submission pdf accordingly.
> > This method still requires existinig region proposal generation models, like ground-truth, RPN, GLIP, which makes this method extremely limited. From this view, this method is even not complete, since you cannot find such a region understanding task in reality. In addition, the pre-extracted regions are also meaningless, since SAM can also perform the region proposal generation task. Therefore, the main experimental setting is meaningless and unreasonable.
>
> Appologies for the confusion. We highlight the following points:
>
> 1. Same as our work, several recent works such as RegionCLIP and DetPro also use extrenel region proposal, as we all focus on recognizing the region of interest. As showin in ViLD and F-VLM, existing region proposals methods are already highly generic across different domains, whilst recognizing the detected regions is relatively outpaced. That's why we focus on the latter.
> 2. Our method fully preserves SAM flexible prompting capability, which enables RegionSpot to perform interactive region recognition, extract region-specific semantic features, and object localization.
> 3. Proposal generation from SAM: Whilst SAM can generate region proposals, its recall is still significantly inferior than other alternatives such as GLIP, as shown in the table below, leading to less competitive final results.
>
> | Proposal | AP_rare | Recall_200 |
> | --- | --- | --- |
> | SAM | 10.6 | 0.4 |
> | RPN | 10.9 | 0.53 |
> | GLIP | 20.0 | 0.75 |
>
>
> > There are also many alternatives to perform the so-called region understanding task with SAM and CLIP. For example, SAM can directly extract region proposals and CLIP (or RegionCLIP) can predict the category tags of them. We can also add some projection layers in this way, freeze most parameters and finetuning a small part of parameters for efficient training. The author should perform experiments to compare with baselines like this. Otherwise, the effect of the position-aware localization cannot be seen.
>
> Thanks for such detailed suggestions. In our submission, we already provided two variants of this suggested baseline by feeding ground-truth proposals in the first two rows of Table 1. They can be considered as the upper bounds. Following the suggestion, we further tested with SAM region proposals along with CLIP. The table below shows simply streamlining the off-the-shelf foundation models is still largely inferior than our method. We have added this new baseline in the revised version.
>
> |  | Proposals | AP_rare |
> | --- | --- | --- |
> | SAM + CLIP | SAM | 8.6 |
> | SAM + CLIP w/ projector | SAM | 9.6 |
> | RegionSpot-BB | SAM | 10.6 |
> | RegionSpot-BB | GLIP | 20.0 |
>
>
> > The experiment in Table 1 is also unfair. RegionCLIP simply performs image-text pre-training, without object detection finetuning. However, RegionSpot performs something about detection training. Therefore, the author should finetuning RegionCLIP on the same detection datasets for a fair comparison.
>
> Thank you for the excellent suggestion. Given that RegionCLIP requires extensive computational resources and time for re-training, we conducted experiments following the ViLD protocol for a fair comparison. The results, presented in Table 2 (Response to all reviewers), demonstrate that our method achieves a 19.7%, outperforming RegionCLIP 17.1% by 2.6% APr. This is noteworthy as we achieve these results without refining the backbone using large-scale image-text pairs.
>
> > The author should also compare with some more recent methods, like GLIP v2, Grounding DINO and so on.
>
> Thank you for the valuable suggestion. As GLIP v2 does not operate in a zero-shot manner on LVIS, we have included the more robust Grounding DINO large model in Table 1 (Response to all reviewers) for a direct comparison. Please refer to this for detailed insights.

---

> ### Author Response · Authors · 2023-11-22
> **Last discussion request before the discussion period ends**
>
> Dear Reviewer，
>
> Thank you for dedicating your time and expertise to review our paper and for participating in the rebuttal process. Your feedback has been instrumental in enhancing the quality of our work.
>
> We hope that our responses in the rebuttal have satisfactorily addressed your concerns. If there are any remaining issues or new points of discussion, we are fully prepared to engage in further dialogue. Given the constraints of the review timeline, we would greatly appreciate it if you could review our revised responses at your earliest convenience.
>
> Should you find that our revisions and clarifications have resolved the initial concerns, we would be grateful for your reconsideration of the initial rating. However, if there are still aspects that require clarification, we welcome the opportunity to discuss them in the remaining time.
>
> We sincerely appreciate your time and thoughtful consideration.

---

### Official Review · Reviewer_UsjU · 2023-11-01

**Soundness:** 3 good
**Presentation:** 4 excellent
**Contribution:** 3 good
**Rating:** 6
**Confidence:** 4

**Summary:**

Authors propose RegionSpot, an open-vocabulary detection and instance segmentation approach that leverages frozen foundation models. Specifically, given some set of candidate bounding box proposals, authors use SAM for class-agnostic localization and CLIP features for classification. Importantly, authors only train a small projection and attention module to combine the location queries from SAM with the semantic key/value pairs from CLIP. Authors evaluate their method on LVIS and find that their method beats prior work including RegionCLIP and GLIP.

**Strengths:**

- Simple Approach. Authors propose a simple way of combining two popular off-the-shelf foundational models for vocabulary detection and segmentation that effectively leverages the foundational pre-training of each model.
- Excellent Training and Data Efficiency. Due to the small size of the RegionSpot attention module, authors can train on 3M data points in 22 hours. (Table 2 is notable)
- Clear Explanation. Authors present their work in a generally coherent manner.

**Weaknesses:**

- Limited Baseline Comparisons. Despite significant prior work in OVOD [1,2,3], authors primarily only compare with RegionCLIP and GLIP. In reality, prior work [1] significantly outperforms RegionSpot.
- Unfair Comparisons. Since authors show that pre-training data scale significantly contributes to model performance, comparing RegionSpot, a technically very similar method trained on much less data is unfair. Instead, it would make more sense to evaluate RegionSpot trained on only CC3M.
- Limited by Quality of Boxes. RegionSpot is always limited by the bounding box proposals provided as input to the system. As authors show in Table 1, the type of proposals has a significant impact on model performance. It would be interesting to evaluate how the impact of using proposals from one of the more performant open-vocabulary models than GLIP (e.g. GroundingDINO).

References

[1] Scaling Open-Vocabulary Object Detection. Minderer et. al. ArXiv.

[2] Multi-Modal Classifiers for Open-Vocabulary Object Detection. Kaul et. al. ICML 2023

[3] https://github.com/witnessai/Awesome-Open-Vocabulary-Object-Detection

**Questions:**

- How to Deal with False Positive Boxes? Since all regions are classified into one of K categories, how are false positive proposals addressed?
-  Can this method be most improved by better classification (e.g. CLIP) or localization (e.g. SAM)? What are the typical error modes?

---

> ### Author Response · Authors · 2023-11-20
> **Official Response to Reviewer UsjU**
>
> Thank you for the helpful review. We carefully address your questions and comments below and have updated the submission pdf accordingly.
> > Limited Baseline Comparisons. Despite significant prior work in OVOD,  authors primarily only compare with RegionCLIP and GLIP. In reality, OWL-V2 significantly outperforms RegionSpot.
>
> Thanks for pointing this out. We note that comparing OWL-V2 and RegionSpot directly is not fair in terms of the usage of data scale: WebBI dataset at a size of 2B images vs. 3M images. This big difference in data size makes us hard to draw meaningful insights.Following additional suggestions, we have included more extensive comparison. We add  GLIP-L and Grounding DINO-L in Table 1 ( Response to all reviewers) to verify the efficient training and good performance.
>
> > Unfair Comparisons. Since authors show that pre-training data scale significantly contributes to model performance, comparing RegionSpot, a technically very similar method trained on much less data is unfair. Instead, it would make more sense to evaluate RegionSpot trained on only CC3M.
>
> Our training objectives differ from those of RegionCLIP, which requires large-scale image-text pairs to retrain a region-level backbone. Our goal is to leverage publicly available object detection datasets to train new layers that enable existing foundational models to perform region-level understanding. For fair comparison, we have conducted experiments following the ViLD protocal Please refer to our comprehensive response to all reviewers for detailed results.
> > Limited by Quality of Boxes. RegionSpot is always limited by the bounding box proposals provided as input to the system. As authors show in Table 1, the type of proposals has a significant impact on model performance. It would be interesting to evaluate how the impact of using proposals from one of the more performant open-vocabulary models than GLIP (e.g. GroundingDINO).
>
> Thanks for this insight. We have now evaluated the effect of input proposals on performance. As GroundingDINO comes with no open-source for LVIS evaluation, we instead utilize GLIP-L as the proposal generator. Our results underscore two key findings:
> 1. There is a significant improvement when the quality of prompt boxes is improved.
> 2. Despite GLIP-L generating high-quality boxes, it faces challenges in region recognition. Notably, RegionSpot achieves superior performance, exceeding GLIP-L by 13.1 in AP_r metric.
> |  | Proposal | AP_rare |
> | --- | --- | --- |
> | GLIP-L | GLIP-L | 17.1 |
> | RegionSpot-BL | GLIP-T | 24.9 |
> | RegionSpot-BL | GLIP-L | 30.2 |
>
>
> > How to Deal with False Positive Boxes? Since all regions are classified into one of K categories, how are false positive proposals addressed?
>
> We treat the background as a special class, which aids in mitigating issues associated with false positives.
>
> > Can this method be most improved by better classification (e.g. CLIP) or localization (e.g. SAM)? What are the typical error modes?
>
> Thanks for the great question. We would do more evaluation on their respective effects once more options become available. From our (not exhaustive) observation, it is hard to draw the typical error modes though.

---

### Official Review · Reviewer_PEMU · 2023-11-10

**Soundness:** 2 fair
**Presentation:** 2 fair
**Contribution:** 2 fair
**Rating:** 5
**Confidence:** 3

**Summary:**

This paper focuses on the challenging task of region recognition, particularly, open-world object detection. The authors present a new region recognition framework, combining a vision foundation model, i.e., SAM, and a vision-language foundation model, i.e., CLIP. In this framework, localization knowledge and semantic conception are integrated to promote each other. Experimental results and analyses conducted on the LVIS detection dataset demonstrate its effectiveness and generalization.

**Strengths:**

**Originality**: The paper proposes a lightweight knowledge integration module to unleash the ability of both SAM and CLIP models for open-world object detection tasks.

**Quality**: The paper provides a thorough experimental evaluation of **RegionSpot** on a challenging object detection dataset, i.e., LVIS. The authors also conduct various ablation studies to analyze the impact of different components of **RegionSpot**, such as the scale of training data, different representations from the CLIP model, selection of position-aware tokens, etc.

**Clarity**: The paper also provides sufficient background information and related work to situate the contribution of **RegionSpot** in the context of existing literature on region understanding and zero-shot recognition.

**Weaknesses:**

**Major Issues**:

**Insufficient novelty and contribution**: The newly proposed **RegionSpot** framework lacks justification for its design. The pipeline of fine-tuning a lightweight module while frozen SAM and CLIP models seems natural and basic. Additionally, only conducting experiments on object detection tasks is not convening.

**Insufficient results for experiments**: Although the authors claim that "This implementation effectively facilitates the fusion of semantic and location information in a manner that is amenable to learning and yields substantive efficacy.", they provide no experimental results. Also, the motivation is not clear. For example, why do the authors serve the localization feature as the role of `query`? what if the ViL feature assumes as the `query`?

**Minor Issues**:

**Excessive statement**: The authors claim that "Our model's flexible architecture allowed us to seamlessly replace one-hot labels with class name strings.". This may be overemphasizing their contribution.

**Grammar and minor errors**:
- In section **Position-aware tokens selection in SAM**, a grammatical error is present in the sentence: "Surprisingly, although it can outperform GLIP (i.e., 17.2 vs. 18.6)."
- In section **Prompt enginerring**, there is a discrepancy in the reported increase: "an increase of 1.4 AP" — should this be 1.6 AP? Additionally, clarification is needed regarding whether the method in the last line of Table 5 indeed represents the baseline with both prompts.

**Questions:**

1. My major concern is the contributions of combining SAM and CLIP for object detection tasks.

2. The authors should discuss the limitations and potential negative societal impact in the Conclusion.

3. Please also refer to Weaknesses.

---

> ### Author Response · Authors · 2023-11-20
> **Official Response to Reviewer PEMU**
>
> Thank you for the helpful review. We carefully address your questions and comments below and have updated the submission pdf accordingly.
>
> > The newly proposed RegionSpot framework lacks justification for its design.  Although the authors claim that "This implementation effectively facilitates the fusion of semantic and location information in a manner that is amenable to learning and yields substantive efficacy.", they provide no experimental results. Also, the motivation is not clear. For example, why do the authors serve the localization feature as the role of query? what if the ViL feature assumes as the query?
>
> 1. Our motivation is to integrate position-aware localization knowledge from a localization foundation model (e.g., SAM) with semantic information extracted from a ViL model in a frozen to achieve the region understanding.  As Reviewer Tjjg  said, introducing vision foundation models into new tasks is an important thing.
> 2. In our framework, the mask tokens from SAM, which act as queries, inherently localize objects but do not convey semantic information. Our method thus uses these tokens as query to find the corresponding semantic details from the ViL feature map, enhancing the semantic understanding at a regional level. Using the ViL feature as a query, on the other hand, would be illogical for our purpose, as our objective is to augment already localized specific regions with semantic information from the ViL features.
>
> > Only conducting experiments on object detection tasks is not convening.
>
> As we focus on region recognition, object detection is the most important and most suitable problem for evaluation in computer vision, per our knowledge. Also, we have evaluated the Instance Segmentation task with positive comparisons. We are open to evaluate more suitable problems if suggested kindly.
>
> > Overemphasizing the contribution about "Our model's flexible architecture allowed us to seamlessly replace one-hot labels with class name strings."
>
> We would tune done and refine this description. To further clarify: Using text embedding of class name strings instead of one-hot labels is the enabler for good zero-shot performance as exemplified with recent foundation models like CLIP and ALIGN (both function at the image level). In the domain of open-world object recognition, this approach is similarly employed to attain zero-shot capabilities, such as ViLD and GLIP. We have simply adhered to this established method.
> > Ablation Study of Prompt engineering - "clarification is needed regarding whether the method in the last line of Table 5 indeed represents the baseline with both prompts."
>
> Yes, we will clarify in the revision.
> > The authors should discuss the limitations and potential negative societal impact in the Conclusion.
>
> Thank you for emphasizing the need to address limitations and potential negative societal impacts in our Conclusion. We recognize that our method, while offering advancements in open world region understanding, has limitations, including its dependency on external region proposal mechanisms. This could potentially limit its versatility or introduce biases depending on the proposal generation source. To achieve more efficient AI, there is good amount of space to compress our whole architecutre by using smaller faster-to-run compoents whilst paying little cost in recognition performance. Moreover, we will discuss possible negative societal impacts, such as concerns over privacy, the ethical use of recognition technologies, and the imperative for responsible, transparent deployment of such tools. Your feedback is invaluable in ensuring our research comprehensively addresses these critical and broader implications.
>
> > Grammar mistakes
>
> Thanks, we will fix all in the revision.

---

> ### Author Response · Authors · 2023-11-22
> **Last discussion request before the discussion period ends**
>
> Dear Reviewer，
>
> Thank you for dedicating your time and expertise to review our paper and for participating in the rebuttal process. Your feedback has been instrumental in enhancing the quality of our work.
>
> We hope that our responses in the rebuttal have satisfactorily addressed your concerns. If there are any remaining issues or new points of discussion, we are fully prepared to engage in further dialogue. Given the constraints of the review timeline, we would greatly appreciate it if you could review our revised responses at your earliest convenience.
>
> Should you find that our revisions and clarifications have resolved the initial concerns, we would be grateful for your reconsideration of the initial rating. However, if there are still aspects that require clarification, we welcome the opportunity to discuss them in the remaining time.
>
> We sincerely appreciate your time and thoughtful consideration.

---

### Author Response · Authors · 2023-11-20
**Response to all reviewers**

We appreciate all the reviewers for constructive comments and suggestions. We are encouraged reviewers consider our proposed method novel and good (Reviewers Tjjg , Reviewers UsjU) idea with good performance (Reviewers PEMU, Reviewers Tjjg, Reviewers UsjU) and efficent training cost and data (Reviewers 3vj6, Reviewers UsjU)
 > Evaluation under GLIP protocol

Table 1: Comparison under GLIP protocal.
| Method | Training Data | Data Size | Fully finetune | Training Time | MiniValAPr | MiniValAPall | Val APr | Val APall |
| --- | --- | --- | --- | --- | --- | --- | --- | --- |
| RegionCLIP | CC3M | 3M | Yes | 4.6K | - | - | 13.8 | 11.3 |
| GLIP-L | FourODs,GoldG,Cap24M | 27M | Yes | 120K | 28.2 | 37.3 | 17.1 | 26.9 |
| Grounding DINO-L | O365,OI,GoldG,Cap4M,COCO,RefC | 8M | Yes | - | 22.2 | 33.9 | - | - |
| RegionSpot-BL | O365,OI,V3DET | 3M | **No** | **0.2k** | **33.2** | 36.9 | **30.2** | **29.8** |


 > Evaluation under ViLD protocol.

Table 2: Comparison under ViLD protocal.
|  | Proposals | AP_Rare |
| --- | --- | --- |
| ViLD | RPN | 16.1 |
| RegionCLIP | RPN | 17.1 |
| Detic-ViLD | RPN | 17.8 |
| F-VLM | RPN | 18.6 |
| RegionSpot | RPN | **19.7** |
| RegionSpot | GLIP-T | **20.4** |

**1. More Comparsion**

As suggested, we add strong baseline (i.e. GLIP-L and Grounding DINO-L) and ViLD protocol to further demostrate our method. For the GLIP protocol setting, we use output boxes of  GLIP-L as our region prompts to evluation the region recognition ability.  As shown in the Table 1, although we use less 9x training data and 600x training time, our models maintains superior performances over their GLIP-L by 13.1 APr on the LVIS 1.0 and 5.0 APr on the MiniVal.  Further, our method also surpasses Grounding DINO-L 11.0 APr on the MiniVal, which used the advanced detector.

To make a more comprehensive evaluation of our method, we also perform experiments under the ViLD protocol, i.e., the method is trained on base categories and then evaluated on novel categories using the original LVIS AP metric not using extra data. The results are shown in Table 2 that our method (19.7 APr) outperforms F-VLM (18.6  APr) by 1.1% APr, which is also built upon Frozen Foundation model. It can reveal our model can fully unleashing the pretrained knowledge.

**2.Fair Comparison with RegionCLIP**

Without the need for backbone training/retraining/fine-tuning, our method is more efficient in terms of both training data and compute resource. Only object detection data is needed to train new learnable layers.For fair comparison with RegionCLIP, we evaluate under the ViLD protocol. RegionSpot was just trained with LVIS base class. The results in Table 2 show  that our method (19.7 APr) outperforms RegionCLIP (17.1 APr) by 2.6% APr despite NOT refining the backbone with large scale image-text pairs.

---

### Author Response · Authors · 2023-11-20
**Paper Revision**

We express our sincere gratitude to all reviewers for their constructive and insightful comments. Your suggestions have been invaluable in enhancing the quality of our paper. We have carefully revised the manuscript, incorporating your feedback.

Detailed Updates:

**1. Stronger Baseline Experiments under GLIP Protocol:** Added GLIP-L and GroundingDINO-L experiments in Tables 1 and 7.

**2. Comparison under ViLD Protocol:** Conducted additional experiments under the ViLD protocol for a more comprehensive evaluation, as shown in Table 8.

**3. Ablation Study of SAM-CLIP Pipeline:** Included experiments on several alternatives integrating SAM and CLIP in Table 9(a).

**4. Ablation Study of SAM Model Size:** Added an ablation study on SAM model sizes in Table 9(b).

**5. Model Details:** Updated Table 2 with information on learnable parameters.

**6. Correction of Typos and Reference Errors:** Thanks to Reviewers PEMU and 3vj6 for pointing out writing errors, which we have now corrected.

We thank all reviewers again for their valuable suggestions, which have significantly contributed to the improvement of our paper. We welcome any further concerns and suggestions and remain committed to enhancing our work.

---

### Meta-Review · Area_Chair_Y8YY · 2023-12-04

**Metareview:**

This paper receives 3 negative reviews and 1 positive review initially. The raised issues include insufficient novelty and contribution, insufficient experiments, unfair comparisons, existence of motivation on local region understanding, unclear experimental configurations and statistics. During rebuttal, the authors try to address these issues by providing additional experiments, emphasizing the design of object detection, and clarify technical details. Overall, the AC has checked all the comments, and stands on the reviewers' side that the current work lack sufficient contributions. The main motivation is to align frozen VLM to enrich their attention on the local image region. This alignment is via a projection module while keeping the other parts fixed. This design shares many similarities with visual grounding, RegionClip, and DetClip. The authors are suggested to carefully elucidate the difference upon existing works, especially from the object detection perspective, to strengthen the proposed contribution. Welcome for the next venue.

**Justification For Why Not Higher Score:**

Not enough contribution as pointed out by reviewers.

**Justification For Why Not Lower Score:**

N/A

---

### Decision · Program_Chairs · 2024-01-16

Reject